# Supporting grant reviewers through the scientometric ranking of applicants

**Balázs Győrffy**[1,2,3]*, **Boglarka Weltz**[1,3], **István Szabó**[4]

**1** Department of Bioinformatics and 2nd Department of Pediatrics, Semmelweis University, Budapest, Hungary, **2** RCNS Cancer Biomarker Research Group, Institute of Enzymology, Budapest, Hungary, **3** National Laboratory for Drug Research and Development, Budapest, Hungary, **4** Budapest Business School, Budapest, Hungary

* gyorffy.balazs@med.semmelweis-univ.hu

## Abstract

### Introduction

Comparing the scientific output of different researchers applying for a grant is a tedious work. In Hungary, to help reviewers to rapidly rank the scientific productivity of a researcher, a grant decision support tool was established and is available at www.scientometrics.org. In the present study, our goal was to assess the impact of this decision support tool on grant review procedures.

### Methods

The established, publicly available scientometric portal uses four metrics, including the H-index, the yearly citations without self-citations, the number of publications in the last five years, and the number of highly cited publications of a researcher within eleven independent scientific disciplines. Publication-age matched researchers are then ranked and the results are provided to grant reviewers. A questionnaire was completed by reviewers regarding utilization of the scientometric ranking system. The outcome of the grant selection was analyzed by comparing scientometric parameters of applying and funded applicants. We compared three grant allocation rounds before to two grant allocation rounds after the introduction of the portal.

### Results

The scientometric decision support tool was introduced in 2020 to assist grant selection in Hungary and all basic research grant applicants (n = 6,662) were screened. The average score of funded proposals compared to submitted proposals increased by 94% after the introduction of the ranking. Correlation between ranking scores and actual grant selection was strong in life and material sciences but some scientific panels had opposite correlation in social sciences and humanities. When comparing selection outcome to H-index across all applicants, both type I and type II errors decreased. All together 540 reviewers provided feedback representing all eleven scientific disciplines and 83.05% of the reviewers (especially younger reviewers) found the ranking useful.

**Data Availability Statement:** All relevant data are within the paper.

**Funding:** The author(s) received no specific funding for this work.

**Competing interests:** The authors have declared that no competing interests exist.

## Conclusions

The scientometric decision support tool can save time and increase transparency of grant review processes. The majority of reviewers found the ranking-based scientometric analysis useful when assessing the publication performance of an applicant.

## Introduction

Research and development (R&D) represents systematic creative activity done in the area of science and technology with the purpose of increasing the level of knowledge [1]. Worldwide, 1.3 million scientific documents were published in 2000 –this number almost tripled to 3.7 million in 2020. A country's R&D performance can be measured in various ways. The SCImago Country Rank (http://www.scimagojr.com) is a public database providing journal- and country-specific publication measures. SCImago is based on the Scopus database and provides a ranking for all scientific journals by assigning them into four quartiles where Q1 is the best and Q4 is the weakest cohort. It also contains aggregate data useful for comparing the scientific output of different countries. Decision makers and also the broader R&D community employs in many cases some other figures, like the GERD (Gross domestic Expenditure on R&D), the total number of publications, the number of researchers, or the number of patents.

What is the scientific position of Hungary according to these major indicators? In terms of R&D spending, Hungary is standing in a decent position, not far from the EU average (1.61% in Hungary vs 2% EU average according to Eurostat). According to the Hungarian Central Statistical Office, the total number of researchers shows a steady increase, from 53,911 researchers in 2010 to 73,412 researchers in 2020. However, when analyzing SCImago data for Hungary, a steady drop in ranking position can be observed in the last two decades (**Fig 1**). Although the total number of publications grows, the pace of scientific output increase lags behind similar EU countries.

A country's performance is the sum of the outputs of all national researchers. Therefore, on the level of individual researchers, especially in the case of grant schemes, which provide funding based on researcher quality, it is of utmost importance that the supported projects should result in excellent and frequent publications wherever possible. The National Research, Development and Innovation Office (NRDIO), as the main funding organization for individual researchers' projects in Hungary is working closely with more than 5,000 reviewers for its grant applications. This Office introduced a system that shows scientific performance based on the current international standards, which motivates both the reviewers and the researchers to focus even more on research impact. According to the Leiden Manifesto, "decision-making about science must be based on high-quality processes that are informed by the highest quality data"[2]. With this in mind, the NRDIO adopted a reviewer support tool (www.scientometrics.org) that shows the reviewers the individual applicants performance based on currently available metrics. Notably, the implemented system differs from available tools generally used for the evaluation of scientific output (like the H-index, citation count, or number of publications), because it provides an age-matched ranking. The age-matched ranking can provide similarly high scores for younger and older researchers. The tool's solitary ranking is based on the concept that "simplicity is a virtue in an indicator because it enhances transparency" [2].

In the current study, two years after its introduction, our goal was to assess the impact of this decision support tool on grant review procedures. The tool is assembled based on the results of our previous studies, in which we analyzed individual researchers' output in relation to scientometric parameters and reviewer scores [3], to discipline-specific past performance [4], and to

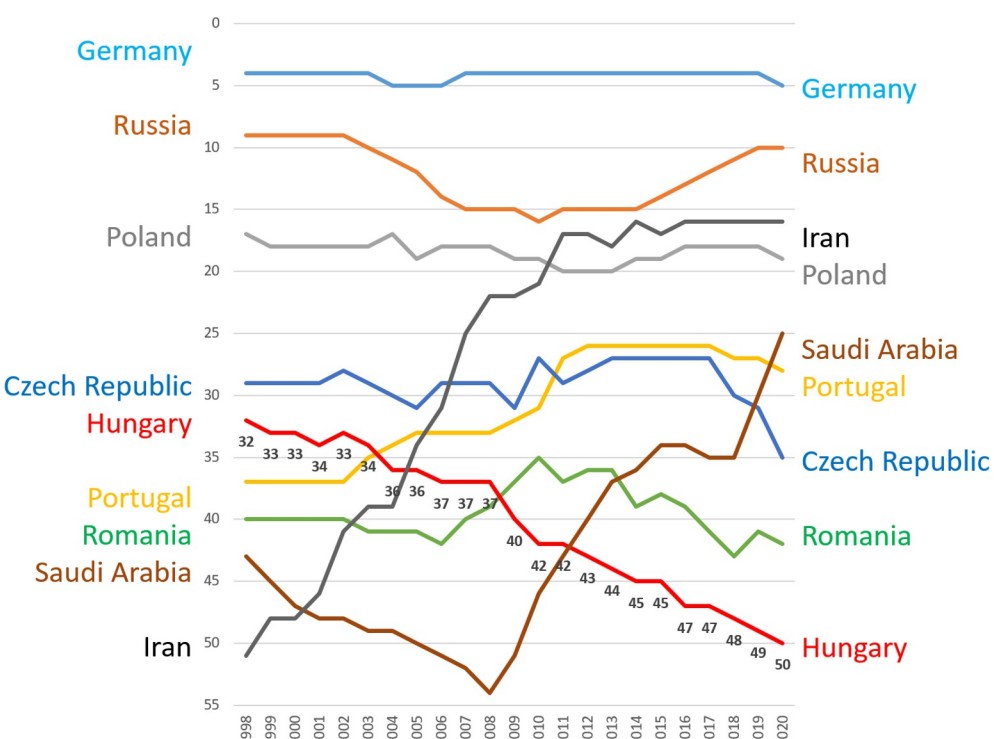

**Fig 1. Selected countries in the SCImago country rank index compared to Hungary.** While most Eastern European countries were able to maintain their overall ranking, Hungary consistently reached lower scores in the last twenty years. Some countries were able to achieve significant growth.

the different age of the scientists [5]. In the present study, we aimed to determine the effects of the established decision support tool on applicant selection and on the reviewers' satisfaction.

## Methods

### Publication and citation data

Publication and citation data for Hungarian researchers was obtained from the Hungarian Scientific Work Archive (MTMT, www.mtmt.hu). This user-maintained repository comprises all research papers, books, proceeding, and other publications of Hungarian researchers as well as all citations these received. Of note, unlike other sources like Google Scholar, the citation data in MTMT is catalogued to list self-citations and citations without self-citations for each publication. Self-citation occurs in case there is an overlap between the author lists of the cited and the citing document. Birth year of the researchers was downloaded from the doktori.hu database.

### The scientometrics.org system

The scientometrics.org platform uses three principal parameters to determine scientometric output for a selected researcher. These include the H-index, the number of citations received in the last complete calendar year, and the number of publications in the last five calendar years (**Fig 2A**). The H-index is determined using all citations received for all publications. For the citation parameter, only citations without self-citations received in the last calendar year are included for all publications. The number of publications is determined differently in each

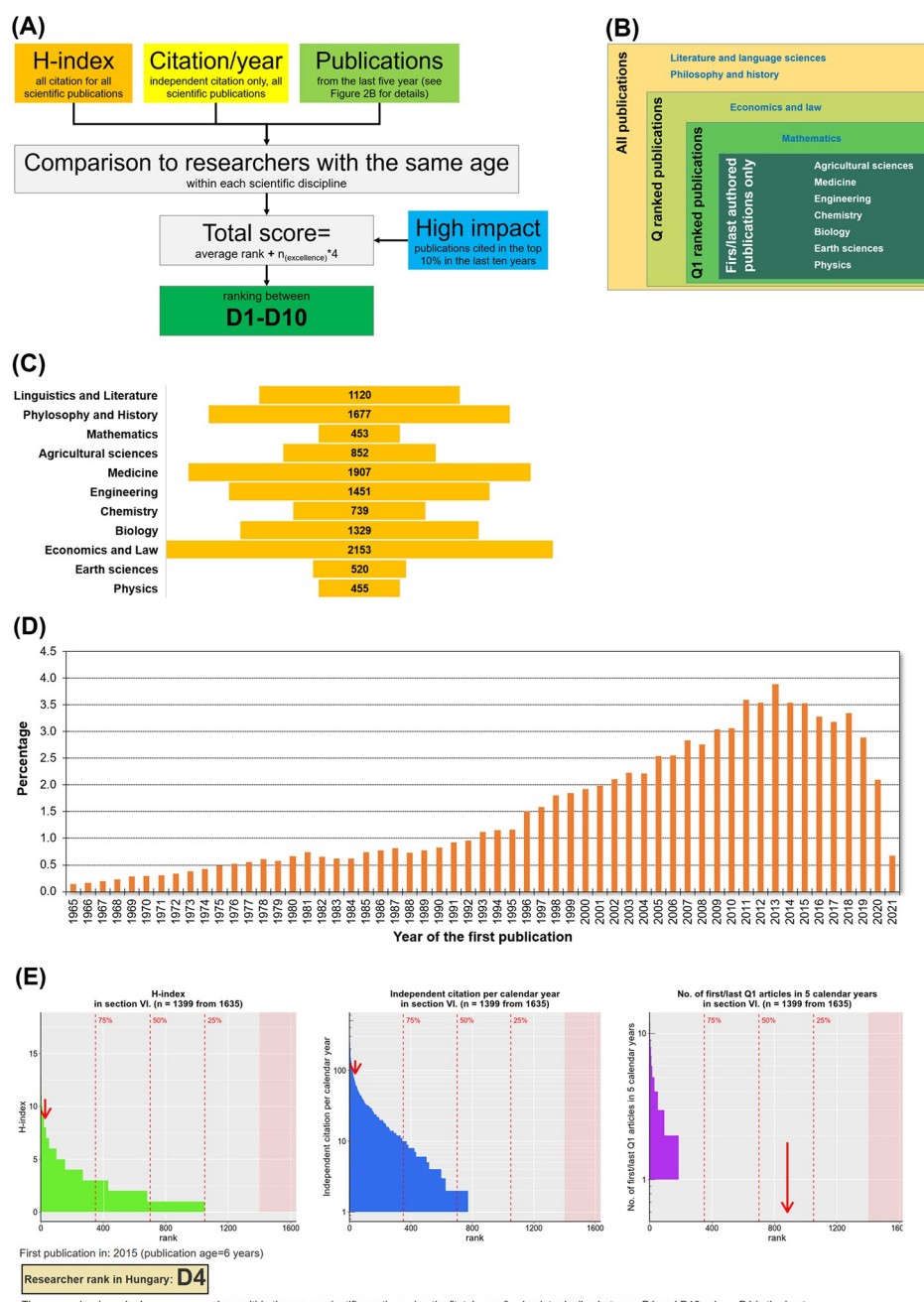

**Fig 2. Setup of the scientometrics.org decision support tool.** The included parameters (A), the scientific discipline-specific inclusion of publications (B), the distribution of researchers across the different scientific disciplines in the reference database (C), the distribution of age across the entire database (D), and representative example of the provided ranking for a randomly selected researcher (E).

scientific discipline: all publications (e.g. journal articles regardless of ranking, books, book chapters, and abstracts) are counted in humanities, only Q ranked papers are considered in economics, only Q1 ranked publications are used in mathematics, and only first- or last-authored Q1 ranked publications are utilized in all other disciplines (**Fig 2B**). The disciplines were set up using the classification of the Hungarian Academy of Sciences as described in our

previous publication [5]. For each parameter, the absolute values were computed for each researcher for each year. The initial year is set as the year of the first publication, and the number of years passed since this year is termed the researcher's "publication age". Finally, researchers are ranked across researchers with the same publication age within the same scientific section into percentiles. Ties are handled by using the median rank for each included researcher. To increase simplicity, the average of the three percentiles is computed (in this, the number of publications has a double weight), and this final score is transformed into deciles between D1 and D10, where D1 is the best.

The web-based scientometric decision support portal was adopted in 2020 to assist grant selection and is available at www.scientometrics.org (**Fig 2C**). A snapshot of the system is deposited at www.tudomanymetria.com. The tudomanymetria.com site is only updated once a year, right after the yearly deadline of the basic research grants. This enables the reproducible comparison of researchers without the influence of new publications or citations attained after the grant cutoff date.

### Reviewer feedback

With one and a half thousand grant applications evaluated yearly, the basic research grant of the Hungarian Scientific Research Fund is the most widely distributed grant available in Hungary. Since 2020, all basic research grant applicants of the Hungarian Scientific Research Fund are screened by using the scientomterics.org algorithm and the results is provided to grant reviewers in a table format.

Last year, a web questionnaire was conducted after completing the yearly review round for basic research grants. The questionnaire was sent to all reviewers electronically. In addition to the general data including position and scientific discipline, an assessment of the decision support tool was requested. Reviewer responses were collected anonymously. Ethical approval for the research was granted by the Institutional Ethical Board under the number NKFIH-2729-6/2020.

### Statistical analysis

A detailed description of the statistical pipeline for determining a researcher's total score is provided above. During the system setup we assessed different parameters and used Spearman rank correlation to evaluate the relationship between continuous variables (e.g. publication age and biological age). Percentiles were used to rank and compare the total score values of submitted and funded applications. Median and mean of all total score values in different cohorts were computed to measure the magnitude of the difference. Statistical significance when comparing continuous variables between submitted and funded proposals was determined using Kruskall-Wallis or Mann-Whitney tests. Type I errors and type II errors were computed as described previously [6]. The cutoff for statistical significance was set at $p<0.05$. Country ranking data was obtained from the SCImago database (https://www.scimagojr.com/). Descriptive usage statistics data for website visitor trends was obtained using the Universal Analytics module available in Google analytics (https://analytics.google.com).

## Results

### The decision support tool

The scientometrics.org decision support is currently available for 41,448 researchers. Ranking is based on 17,071 researchers who declared completeness of their MTMT record. Of these, 1,427 belong to language and literature (8.36%), 2,266 to philosophy and history (13.27%), 599 to mathematics (3.51%), 1,209 to agricultural sciences (7.08%), 2,476 to medicine (14.50%),

2,191 to engineering (12.83%), 1,102 to chemistry (6.46%), 1,680 to biology (9.84%), 2,732 to economics and law (16%), 693 to earth sciences (4.06%), and 696 to physics (4.08%). The graphical analysis output of scientometrics.org is summarized in **Fig 2C**.

Scientometrics.org was originally developed by using the biological age of the researchers. However, a separate database is needed to keep a record of the birth year for all researchers. For this reason, we tested the utilization of the year of the first publication. The two values had a very high correlation with a Spearman rank correlation coefficient of 0.92 (p<1e-16, n = 16,497). Hence, the year of the first publication was used in all subsequent analyses when determining the researcher's age. The initial version of scientometrics.org also included an option to use the year of the PhD as the reference point. However, this was discontinued because when evaluated by the grant reviewers (see below), over 90% of responders agreed with the cancellation of this feature.

## Utilization in basic research grant evaluation

All together 6,662 basic research grant applications were submitted to the Hungarian Scientific Research Fund between 2017–2021 and 1,960 of these received funding. These numbers translate to a success rate of 29.4% for individual applications. The D ranking as well as the total score values of each applicant were provided for reviewers since 2020.

We computed the total score values for all researchers, including all who submitted their application since 2017. The mean total score of submitted applications was 68.2 and the median score was 76. The mean total score of funded applications before the introduction of the scienomterics.org system (2017–2019) was 76.3 and the median score was 82. After providing the D ranking and the total score values for researchers (2020–2021), the mean total score increased to 82.2 and the median score to 86 (**Fig 3A**).

We computed the difference between the total score values of submitted and funded proposals for each year separately (**Fig 3B**). Overall, the difference in total score values increased by 94% when comparing the difference between submitted and funded proposals before and after the introduction of the scientometric decision support tool (4.81 vs 9.33, p = 0.03). Notably, even before the introduction of the scientometrics.org system, the difference between the submitted and funded total score values had a small (statistically not significant) improvement in each year (**Fig 3B**).

During the five-year period of 2017–2021, 48 applicants received a grant from the Hungarian Scientific Research Fund two times. The average total score value of these researchers was significantly higher than the average total score values of all other funded researchers (77.5 vs. 84, p = 0.0068, **Fig 3C**).

## Type I and type II error

Peer review decisions were previously validated using the H-index as a reference [6]. In this, grant applicants are assigned into four groups. First, those with a H-index over the median ("high") and approved application represent a correct decision. Similarly, those with a low H-index and rejected application also correspond to a correct decision. Funded applicants with a low H-index correspond to a type I error, while rejected applicants with high H-index correspond to a type II error (see **Table 1A**). In this analysis, the number of applicants was normalized to correct for the 26% increase in funding through the 2020–21 period. During the computation of type I and type II errors all applicants were analyzed within their respective scientific discipline, and the final outcomes were aggregated for all researchers. When comparing 2017–2019 grant review rounds to 2020–2021 review rounds, type I error decreased from 10.7% to 8.9% and type II error decreased from 25.5% to 23.7%–for details, see **Table 1B and 1C**.

**(A)**

|  | Submitted | Funded | |
|---|---|---|---|
|  | 2017-2021 | 2017-19 | 2020-21 |
| Minimum | 6 | 7 | 8 |
| 10th Percentile | 43 | 49 | 61 |
| 25th Percentile | 59 | 65 | 76 |
| **Mean** | **68.2** | **76.3** | **82.2** |
| Median | 76 | 82 | 86 |
| 75th Percentile | 88 | 92 | 94 |
| 90th Percentile | 95 | 96 | 97 |

**(B)**

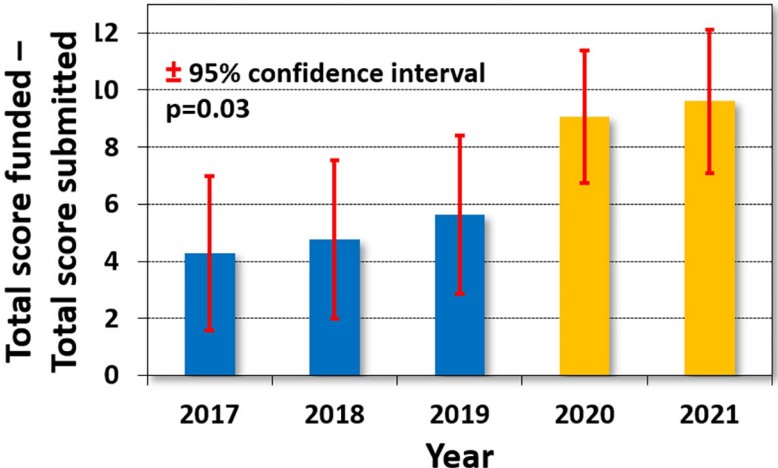

**(C)**

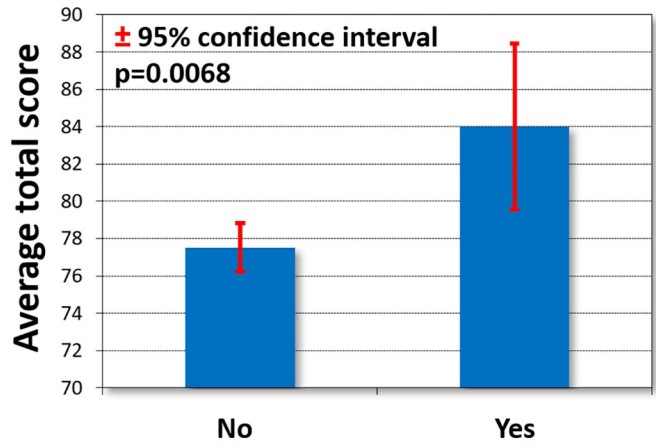

**Fig 3. Total score values of basic grant proposals submitted before and after the introduction of the decision support tool (between 2017–2019 and between 2020–2021, respectively).** Aggregate data for all submitted and funded proposals (A), the difference in the average total score value between submitted and funded proposals in each year (B), and the difference between those who received a single funding vs those who received funding a second time (C).

## Panel differences

The difference between the submitted and funded proposals was not equal across different scientific disciplines. We grouped the different panels into three disciplines (material sciences, life sciences, and humanities and social sciences) and computed the difference between the total score values of submitted and funded proposals in each panel separately. After the introduction of the scientometrics.org ranking, funded proposals had consistently higher total score values (**Fig 4A and 4B**). Although the highest difference between submitted and funded proposals was observed in the psychology and education panel (total score difference = 20.27), in two other panels of humanities and social sciences the funded proposals had markedly lower total score values than submitted proposals. In particular, the total score difference in archeology was -9.28 and in linguistics it was -14.74 (**Fig 4C**).

## Reviewer feedback

Feedback was provided by all together 540 reviewers representing the eleven scientific sections of the Hungarian Academy of Sciences, including language and literature (n = 26, 4.81%), philosophy and history (n = 60, 11.11%), mathematics (n = 19, 3.52%), agricultural sciences (n = 48, 8.89%), medicine (n = 30, 5.56%), engineering (n = 41, 7.59%), chemistry (n = 82, 15.19%), biology (n = 85, 15.74%), economics and law (n = 78, 14.44%), earth sciences (n = 21, 3.89%), and physics (n = 31, 5.74%). The remaining 19 reviewers (3.52%) did not select any of these sections. Notably, the panels of the Hungarian Scientific Research Fund do not exactly overlap with the sections of the Hungarian Academy of Sciences. Nevertheless, we collected the section memberships of the reviewers as one reviewer can be active in multiple panels. According to the occupation, 15.74% of the reviewers were active as senior lecturer (or adjunct / assistant professor, n = 85), 36.11% were employed as associate professor (or docent / adjunct professor, n = 195), 37.96% were full professors (n = 205), and 9.07% had a different job (n = 49). 1.12% of reviewers did not provide response to this question.

**Table 1. Type I error and type II error when comparing the decision outcome and the applicant's H-index before and after the introduction of the decision support tool.** Overview of type I and type II errors (A), proportion of type I and type II errors between 2017–2019 (B) and proportion of type I and type II errors between 2020–2021 (C). For more details about type I and type II errors see [6].

| (A) | Application outcome | |
| --- | --- | --- |
| | **Funded** | **Rejected** |
| **High output** (H-index over median) | Correct | Type II error |
| **Low output** (H-index below median) | Type I error | Correct |
| (B) | Application outcome (2017–2019) | |
| | **Funded** | **Rejected** |
| **High output** (H-index over median) | 13.1% | 25.5% |
| **Low output** (H-index below median) | 10.7% | 50.6% |
| (C) | Application outcome (2020–2021) | |
| | **Funded** | **Rejected** |
| **High output** (H-index over median) | 12.9% | 23.7% |
| **Low output** (H-index below median) | 8.9% | 54.6% |

**(A)**

| Score | Panel | Discipline |
|---|---|---|
| 19.1471 | Earth Sciences 2 | Material sciences |
| 17.9901 | Chemistry 1 | Material sciences |
| 13.6402 | Physics | Material sciences |
| 11.75 | Informatics and Electrical Engineering | Material sciences |
| 11.0333 | Engineering, Metallurgy, Architecture and Transport Sciences | Material sciences |
| 10.2051 | Chemistry 2 | Material sciences |
| 8.0625 | Earth Sciences 1 | Material sciences |
| 5.26667 | Mathematics and Computing | Material sciences |

**(B)**

| Score | Panel | Discipline |
|---|---|---|
| 17.1212 | Plant and Animal Breeding | Life sciences |
| 15.1389 | Clinical medicine | Life sciences |
| 14.7 | Immunity, Cancer and Microbiology | Life sciences |
| 12.1926 | Physiology, Pathophysiology, Pharmacology and Endocrinology | Life sciences |
| 11.9735 | Ecology and evolution | Life sciences |
| 10.95 | Cellular and Developmental Biology | Life sciences |
| 10.2 | omplex Agricultural Sciences | Life sciences |
| 9.72222 | Genetics, Genomics, Bioinformatics and Systems Biology | Life sciences |
| 8.52 | Neurosciences | Life sciences |
| 7.80667 | Molecular and Structural Biology and Biochemistry | Life sciences |

**(C)**

| Score | Panel | Discipline |
|---|---|---|
| 20.2745 | Psychology and Education | Humanities and social sciences |
| 8.27778 | History | Humanities and social sciences |
| 8.15248 | Economics | Humanities and social sciences |
| 7.07143 | Literature | Humanities and social sciences |
| 5 | Law and Government Sciences, Political Science | Humanities and social sciences |
| 2.97778 | Society | Humanities and social sciences |
| 0.25962 | Culture | Humanities and social sciences |
| -9.27778 | Archeology | Humanities and social sciences |
| -14.7436 | Linguistics | Humanities and social sciences |

**Fig 4. The difference between the average total score for submitted and funded proposals.** The applicants were grouped according to scientific panels in material sciences (A), in life sciences (B), and in humanities and social sciences (C).

Altogether, 83.05% of reviewers found the ranking valuable. When asking about the utility of the decision support tool, 90.9% of senior lecturers, 86.2% of associate professors and 77.8% of full professors found the platform useful during grant evaluation process (**Fig 5**).

## Transparency and usage statistics

An important feature of any tool evaluating researcher output is the transparency and reliability of the analysis. To enable reconstruction of the computational steps, the complete publication record of the investigated researcher is computed and provided for download (see **Fig 6** for an example). Besides publication-specific data, the publication record also includes citation data and SCImago ranks for each publication.

Usage statistics for scientometrics.org were recorded in January 2022. According to Google Analytics, the weekly number of visitors in this period was 650 with an average session time of five minutes. Of all visitors, 95.2% came from Hungary confirming that the specificity of the platform is restricted to Hungarian researchers. Daily users / monthly users ratio stood at 9.7% and weekly users / monthly users stood at 47.2% reflecting frequent usage of a smaller user community.

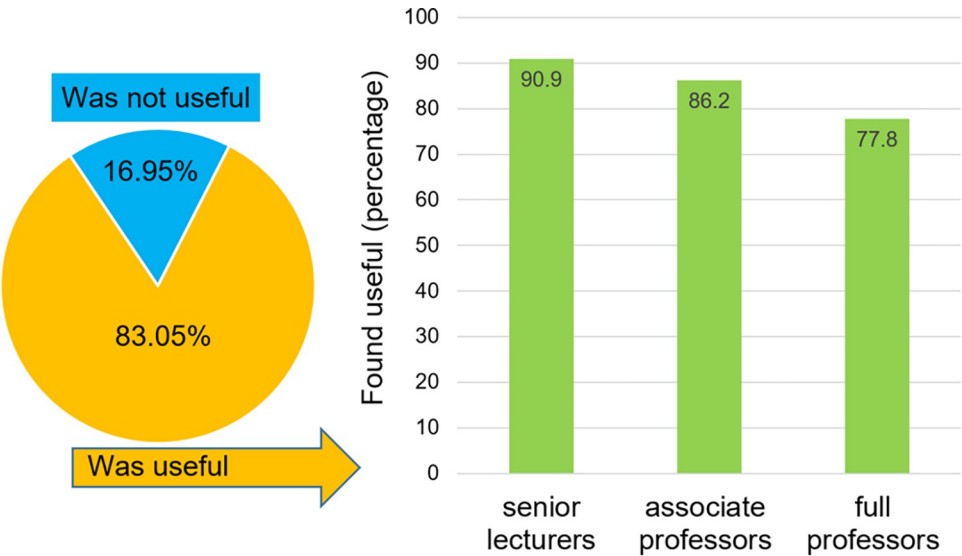

**Fig 5. The majority of reviewers found the ranking-based scientometric analysis useful when assessing the scientometric performance of an applicant.** When stratified by position, a higher proportion of younger researchers (senior lecturers) found the tool useful than older researchers (full professors).

## Discussion

By evaluating 42,905 grant application reviews for 13,303 applications, we have previously exposed a higher publication output for researchers who already had more publications before grant submission [4]. In particular, while reviewers' scores had a minimal correlation with subsequent publication output during the course of the grant time, past scientometric performance of the principal investigator including H-index, citations without self-citations, and number of Q1 publications were the strongest predictors of future output. In another study, we evaluated the scientific output of the Momentum grant scheme and found the highest

|   | A | B | C | D | E | F | G | H | I |
|---|---|---|---|---|---|---|---|---|---|
| 1 | Title | Journal or Publisher | Number of authors | All independent citation | Total citation | Published year | Subtype | First last corresponding author | SJR rating |
| 2 | Multipletesting | PLOS ONE 1932-6203 | 3 | 7 | 7 | 2021 | Research paper | TRUE | D1 |
| 3 | Multi-omics ap | COMPUTATIONAL AND STRUCTURAL BIOTECH | 2 | 21 | 21 | 2021 | Research paper | TRUE | D1 |
| 4 | A személyre sz | SCIENTIA ET SECURITAS 2732-2688 | 4 | 0 | 0 | 2021 | Research paper | TRUE | NA |
| 5 | Gene expressic | CARCINOGENESIS 0143-3334 1460-2180 | 3 | 2 | 2 | 2021 | Research paper | TRUE | D1 |
| 6 | Contingent par | ANIMAL BEHAVIOUR 0003-3472 1095-8282 | 4 | 5 | 6 | 2020 | Research paper | TRUE | D1 |
| 7 | Molecular stra | CANCER AND METASTASIS REVIEWS 0167-765 | 2 | 18 | 18 | 2020 | Review paper | TRUE | Q1 |
| 8 | Network-based | JEDLIK LABORATORIES REPORTS 2064-3942 | 5 | 0 | 0 | 2019 | Abstract (conference proceeding) | FALSE | NA |
| 9 | Alcsoport-spec | MAGYAR ONKOLÓGIA 0025-0244 2060-0399 | 4 | 0 | 0 | 2019 | Review paper | TRUE | Q4 |
| 10 | Gene Expressic | INTERNATIONAL JOURNAL OF MOLECULAR SC | 3 | 8 | 8 | 2019 | Research paper | TRUE | Q1 |
| 11 | Principles of tu | ANNALS OF CLINICAL AND TRANSLATIONAL N | 2 | 6 | 7 | 2019 | Review paper | TRUE | D1 |
| 12 | Molecular mar | JOURNAL OF HEMATOLOGY & ONCOLOGY 17 | 3 | 14 | 15 | 2019 | Review paper | TRUE | D1 |
| 13 | Uncovering Pot | CANCERS 2072-6694 | 6 | 7 | 9 | 2019 | Research paper | TRUE | Q1 |
| 14 | Metaanalízisek | KLINIKAI ONKOLÓGIA 2064-5058 | 2 | 0 | 0 | 2019 | Review paper | TRUE | NA |
| 15 | Mutations Defi | FRONTIERS IN PHARMACOLOGY 1663-9812 | 3 | 13 | 16 | 2019 | Research paper | TRUE | Q1 |
| 16 | The prognostic | BMC CANCER 1471-2407 1471-2407 | 4 | 5 | 5 | 2019 | Research paper | FALSE | Q2 |
| 17 | Determining cc | ROYAL SOCIETY OPEN SCIENCE 2054-5703 | 3 | 159 | 160 | 2018 | Research paper | TRUE | D1 |
| 18 | Demographic s | CLINICAL EPIDEMIOLOGY 1179-1349 | 3 | 12 | 18 | 2018 | Research paper | TRUE | Q1 |
| 19 | Validation of m | SCIENTIFIC REPORTS 2045-2322 | 4 | 783 | 791 | 2018 | Research paper | FALSE | D1 |

**Fig 6. The complete publication record for a researcher as assembled by the scientomteric.org site.** In addition to the title, the name of the journal or publisher (in case of book), the number of authors, the number of citations received, publication year and type, lead authorship, and SCImago journal rank are provided. The table delivers an opportunity to control and validate the computation of the scoring parameters.

correlation between output and the total number of citations, H-index, and the cumulative impact factor in the two most recent years before grant submission [3]. Similar trends of randomness in grant reviewer scores were also documented by other studies in Australia [7] and the United States [8]. Some have even suggested that a modified lottery for research fund allocation could be advantageous compared to the current system [9]. Finally, we explored whether publication characteristics of various scientific disciplines exhibit age-related trends. We have determined a discipline-specific "Golden Age" when the individual scholarly performance peaks [5]. Surprisingly, the results of this study revealed an unexpected degree of predictability with respect to the Golden Age in most analyzed disciplines.

Based on these observations, a publicly available portal was set up to assist Hungarian grant reviewers. The portal compares four scientometric parameters (H-index, number of citations without self-citations per year, publications in the last five years, and the number of high impact publications in the last ten years) and provides a discipline- and publication age normalized ranking for all Hungarian researchers. The ranking values derived by the portal were provided for all reviewers evaluating proposals submitted to the Hungarian Scientific Research Fund in 2020 and in 2021. The reviewers could use the ranking results as additional information when evaluating the CV of the principal investigator and the submitted research plan. Here, we evaluated the grant selection outcome after the introduction of the scientometric ranking and compared the outcome to previous years. Overall, we observed a significantly higher selection of researchers who had a higher publication output before application submission.

Listing the total score value and the ranking value for each applicant can provide important advantages for reviewers. First of all, researchers with negligible publication record and those with exceptional publication output can be identified as the ranking enables grasping the relative performance of a researcher in an effortless and rapid manner. This provides opportunity to focus scarce resources and time on intermediate applications where the research plan in the submitted application can be the primary basis for decision.

Previously, reducing the workload for grant reviewers was the most important recommendation to improve peer review [10]. In line with this expectation, the acceptance of the decision support tool by the peers was very high. All together 83% of reviewers found the provided ranking useful and in each scientific panel (with the exception of two, archeology and linguistics) the ranking had also a strong correlation with the selection. One might ask why these two panels had an opposite correlation. Notably, the highest weight in deriving the score is based on the number of publications. However, sections in humanities and social sciences do not preferentially publish in peer reviewed international journals [11] leading to failed ranking when considering Q ranked publications only. For this reason, we have switched to include all publications, like book chapters, monographs, journal papers, etc. Similarly, H-index values of five, a typical value twenty years after the first publication in these sections, do not allow reliable assessment of publication impact. A future fine-tuning of parameters will be needed to overcome these limitations.

A common criticism of such a system focusing on numerical publication output is that here quantity is put before quality. Available literature data does not support these opinions. For example, a previous project used a Swedish dataset consisting of 48,000 researchers and their WoS-publications to analyze the relation between productivity and production of highly cited papers [12]. The results show that there is not only a robust correlation between productivity (number of publications) and impact (number of citations received), but that this correlation also holds for the assembly of high impact papers: the more papers, the more high impact papers. According to the authors, to write high impact papers, certain output levels seem to be required–of course the numbers depend on which field is under study. Similar results were

obtained in a Canadian study [13]. Here, by using a bulky dataset of disambiguated researchers between 1980 and 2013 (n = 28,078,476), the authors have shown that, on average, the higher the number of papers a researcher publishes, the higher the proportion of these papers are amongst the most cited.

The main function of the described portal is to act as a decision support tool for grant reviewers. A previous European study investigated the predictive validity of grant decision-making, employing a sample of 260 early career grant applications in three social science fields [14]. The authors measured output and impact of the applicants about ten years after grant submission to find out whether the chosen researchers perform ex post better than the non-successful ones. Comparing grantees with non-successful applicants with the best performance, predictive validity was absent. This also suggests that the common belief that reviewers in selection panels are good in recognizing outstanding talents is incorrect. Furthermore, the study also investigated the value of the grants on careers and has shown that grant recipients had a much better career than the non-granted applicants.

With this in mind one could speculate about employing a grant selection solely on the basis of scientometric parameters. However, we cannot suggest this approach. The reason for this is that such analysis tools are prone for optimization [15]. For example, the H-index can be easily leveraged by a few targeted self-citations. Employing certain techniques including collusive and coercive citations can be particularly rewarding when incentives like faculty positions, awards, and grants provide long-lasting motivation for these [16]. A recent paper evaluated the employment and also to the misuse of citation metrics [17]. The study established a homogeneous citation database by using the info of quite 100,000 top scientists. The paper described multiple issues like self-citation, citation-farms, and metrics useful for the identification of unethical citation behavior. For example, citation-farms can dramatically inflate one's scientific output. To enable the investigation of these issues, we have implemented additional parameters including self-citation rate [18], H-index based on self-citation [19], and an algorithm for uncovering citation-farms [17] within the scientometrics.org page.

Scientometrics.org was developed to assess the publication output of individual researchers. As a general rule, a basic research grant of the Hungarian Scientific Research Fund has only one principal investigator. Other researchers can act as participating researchers. Although the website enables the assessment of each individual researcher, there is no option to generate an "aggregate performance assessment" for a group of researchers. A similar task would be the evaluation of entire universities or research institutes. Such aggregate parameters could support the introduction of performance-based funding models for publicly funded research organizations. Such models were previously introduced in several countries like Denmark, Belgium, and Norway [20]. A future extension of scientometrics.org could provide tools for such analyses.

We have to mention a limitation of the utilized approach, as scientometrics can be influenced by internal aspects. As a simple example, in dentistry, if we compare two specialties (e.g. orthodontics [21] versus endodontics [22]), we will have different values. Ultimately, these differences can result in more funding for fields with higher scientometric values. Our use of SCImago journal ranking when determining publication count can help to at least partially abrogate these variances as, even in different specialties, the same proportion of journals will have Q1 rank. A second limitation of the applied methodology is the utilization of a restricted database including Hungarian researchers only: currently, this prevents an assessment on an international scale. Finally, the computed parameters only include citations without self-citations. We have to note, however, that not all self-citations are illegitimate. Including previous papers of the authors in the reference list of a publication can demonstrate the proficiency and expertise of the authors in the topic of the manuscript.

Some initiatives for research assessment in the last two decades evaluate different publication metrics. The most prominent of these are the Open Science Initiative (https://ec.europa.eu/info/research-and-innovation/strategy/strategy-2020-2024/our-digital-future/open-science_en), the Declaration on Research Assessment (DORA, https://sfdora.org/read/), and the Leiden Manifesto [2]. These initiatives share a common view that somehow new evaluation systems should be put in place instead of those that are measuring performance solely on metrics. They seem to dismiss all metrics (at least the "older" ones, like journal rankings and H-index [23]), and would rather resort to the reviewers' opinion in most cases. When metrics appear in their case, most of the time "altmetrics" are mentioned. These incorporate hard to quantify performances as well, like teaching or public outreach. However, to this day, there is no common agreement among researchers on the exact methodology to compute these alternative parameters. At the same time, an altmetric score (a weighted count of all mentions for a publication) has a positive but weak correlation with citation count, at least in health sciences [24]. Until a new research assessment system is in place, the situation in research evaluation is similar to that of the GDP: everybody seems to know and even understand why it's an ill-suited tool for measuring a country's performance, but countries are still judged based upon these [25]. In science, the citations and journal rankings thus have a similar role to that of the GDP-figures when discussing the country's economic performance

In summary, a grant review support tool calculating the scientometric parameters for applicants was introduced in Hungary and here we evaluated the application of this tool after the first two years of usage. With the easily available ranking, a higher proportion of grant applicants with higher scores received funding. When comparing selection outcome to H-index across all applicants, both type I and type II errors decreased after the introduction of the decision support tool. The majority of reviewers (especially younger reviewers) found the ranking-based scientometric decision support tool useful when assessing the scientometric performance of an applicant. One thing can be taken for sure: excellence and impact will remain the cornerstones of researcher evaluation in the future as well, regardless of the alternate methods through which these will be calculated. Therefore, motivating the researchers to publish in a way that it's likely to have an impact and publish in international papers is a strategy that will hold its place in the future. A prospective follow-up of proposals funded after the introduction of the scientometrics.org ranking will provide evidence regarding any potential improvement of the nation-wide publication output of Hungary.

## Acknowledgments

The authors acknowledge the support of Elixir Hungary (www.elixir-hungary.org) and thank Viktoria Lakatos for the careful English editing of the manuscript.

## Author Contributions

**Conceptualization:** Balázs Győrffy.

**Data curation:** Boglarka Weltz.

**Formal analysis:** Balázs Győrffy, Boglarka Weltz.

**Investigation:** Balázs Győrffy, István Szabó.

**Methodology:** Balázs Győrffy, Boglarka Weltz, István Szabó.

**Project administration:** Balázs Győrffy.

**Resources:** István Szabó.

**Software:** Boglarka Weltz.

**Supervision:** Balázs Győrffy.

**Visualization:** Balázs Győrffy, Boglarka Weltz.

**Writing – original draft:** Balázs Győrffy, István Szabó.

**Writing – review & editing:** Balázs Győrffy, Boglarka Weltz, István Szabó.

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
