## [Decision Letter · Decision Letter 0]

24 Oct 2022

PONE-D-22-24996Supporting grant reviewers through the scientometric ranking of applicantsPLOS ONE

Dear Dr. Győrffy,

Thank you for submitting your manuscript to PLOS ONE. After careful consideration, we feel that it has merit but does not fully meet PLOS ONE’s publication criteria as it currently stands. Therefore, we invite you to submit a revised version of the manuscript that addresses the points raised during the review process.

We look forward to receiving your revised manuscript.

Kind regards,

Fausto Cavallaro, PhD

Academic Editor

PLOS ONE

Journal Requirements:

2.Please provide additional details regarding participant consent. In the ethics statement in the Methods and online submission information, please ensure that you have specified (1) whether consent was informed and (2) what type you obtained (for instance, written or verbal, and if verbal, how it was documented and witnessed). If your study included minors, state whether you obtained consent from parents or guardians. If the need for consent was waived by the ethics committee, please include this information.

Reviewers' comments:

Reviewer's Responses to Questions

**Comments to the Author**

1. Is the manuscript technically sound, and do the data support the conclusions?

Reviewer #1: Yes

Reviewer #2: Partly

2. Has the statistical analysis been performed appropriately and rigorously? 

Reviewer #1: Yes

Reviewer #2: Yes

3. Have the authors made all data underlying the findings in their manuscript fully available?

Reviewer #1: Yes

Reviewer #2: Yes

4. Is the manuscript presented in an intelligible fashion and written in standard English?

Reviewer #1: Yes

Reviewer #2: Yes

5. Review Comments to the Author

Reviewer #1: This is an interesting analysis, though limited to one aspect of a grant application.

Would it be possible that the grant assessment process analysed in your study was biased/skewed by the previous dissemination/previous "good name" of the authors? This then reflected on them receiving more funding?

The authors should discuss

what happens if there are several applicants in a single grant application as the values will differ between investigators?

Is the study per se not important when adjudicating research funding?

Some authors make extensive use of social medial, dissemination in other ways. Would it be fair for these to receive more funding?

Would this system skew research funding to those already better known and not allow new researchers to emerge?

Note that scientometrics will be influenced by internal aspects. A simple example, in dentistry, if you compare 2 specialties (e.g. orthodontics versus endodontics, see papers below), you will have different values. Let alone comparing medicine vs dentistry or medicine vs. engineering. Fields with higher scientometric values will gain more funding.

Doğramacı EJ, Rossi-Fedele G. Predictors of societal and professional impact of Endodontology research articles: A multivariate scientometric analysis. Int Endod J. 2022;55(4):312-325. doi:10.1111/iej.13676

Esma J. Doğramacı, Giampiero Rossi-Fedele. Predictors of societal and professional impact of orthodontic research. A multivariate, scientometric approach in Scientometrics (2021)

Reviewer #2: The authors preseted an interesting application of an grant selection decision support system. The anlysi was done competently however the reults are not clearly presented and the discusion is weak. It seems to me that the "exellence" increase based on new metrics provided by the suport tool is quite logical becouse since 2020 reviewers took those metrics into account, and before that their decision was based mostly on the content of the applications. Similar systems/tools are used by quit a few of coutries, so the authors should review them in the introduction and compare them to the Hungarian system in the discusion. They should also criticaly discuss their findings in the disscusion section. The authors shoud also focus more on the question if the review process improved after the tool introduction

6. PLOS authors have the option to publish the peer review history of their article (what does this mean?). If published, this will include your full peer review and any attached files.

Reviewer #1: No

Reviewer #2: **Yes: **Peter Kokol

---

## [Author Response · Author response to Decision Letter 0]

26 Oct 2022

Reviewer #1

> This is an interesting analysis, though limited to one aspect of a grant application.

> Would it be possible that the grant assessment process analysed in your study was biased/skewed by the previous dissemination/previous "good name" of the authors? This then reflected on them receiving more funding?

There is a cohort of 48 applicants who received funding a second time as well. A potential reason for this could be indeed the good reputation of the PI. However, when we compared these applicants to the other grant recipients, their average total score was significantly higher (see Figure 3C) supporting the higher publication output of these researchers.

> The authors should discuss what happens if there are several applicants in a single grant application as the values will differ between investigators?

As a general rule, a basic research grant of the Hungarian Scientific Research Fund has only one principal investigator. 

Other researchers can act as participating researchers. Currently, there is no option to compute an aggregated score for multiple involved researchers or a cohort of scientists. One of our future goals is to expand the tool so that we can derive aggregate values for groups, universities, or research institutions.

We have extended the manuscript Discussion regarding these issues.

> Is the study per se not important when adjudicating research funding?

We thank the reviewer for noticing this very important missing information. 

The grant reviewers received 1) the research plan, 2) the CV of the applicant, and 3) the scientometric ranking provided by the analysis tool. Thus, the selection is not based on the publication parameters only. We extended the manuscript to reference this issue.

> Some authors make extensive use of social medial, dissemination in other ways. Would it be fair for these to receive more funding?

This is another very good suggestion, as there is a positive (but weak) correlation between altmetric scores and citation. We have added this info with a new reference* to the Discussion. As the developed tool does not use on altmetrics, we cannot assess their affect, and as a consequence, we cannot assess whether rewarding these activities could bring additional, measurable benefit for classical citation numbers as well.

* Kolahi J, Khazaei S, Iranmanesh P, Kim J, Bang H, Khademi A. Meta-Analysis of Correlations between Altmetric Attention Score and Citations in Health Sciences. BioMed Research International. 2021;2021: e6680764. doi:10.1155/2021/6680764

> Would this system skew research funding to those already better known and not allow new researchers to emerge?

Better known researchers are generally older. The primary goal of the tool is an age-matched ranking of the researchers. Thus, each researcher is compared to other researchers who have the same publication age. This means, that the very same ranking position can be achieved by a 30 years old and by a 60 years old researcher. This is probably the most important reason why, when stratified by position, a higher proportion of younger researchers found the tool useful than older researchers (see Figure 5.)

> Note that scientometrics will be influenced by internal aspects. A simple example, in dentistry, if you compare 2 specialties (e.g. orthodontics versus endodontics, see papers below), you will have different values. Let alone comparing medicine vs dentistry or medicine vs. engineering. Fields with higher scientometric values will gain more funding.

> Doğramacı EJ, Rossi-Fedele G. Predictors of societal and professional impact of Endodontology research articles: A multivariate scientometric analysis. Int Endod J. 2022;55(4):312-325. doi:10.1111/iej.13676

> Esma J. Doğramacı, Giampiero Rossi-Fedele. Predictors of societal and professional impact of orthodontic research. A multivariate, scientometric approach in Scientometrics (2021)

We extended the Discussion with an additional paragraph describing this issue and we also include the two suggested references.

Reviewer #2

> The authors preseted an interesting application of an grant selection decision support system. The anlysi was done competently however the reults are not clearly presented and the discusion is weak. It seems to me that the "exellence" increase based on new metrics provided by the suport tool is quite logical becouse since 2020 reviewers took those metrics into account, and before that their decision was based mostly on the content of the applications. 

We thank the reviewer for the positive remarks and have improved the Discussion at multiple locations to improve quality.

> Similar systems/tools are used by quit a few of coutries, so the authors should review them in the introduction and compare them to the Hungarian system in the discusion. 

In principle, all grant agencies use some form of evaluation for research output, mostly based on already available data from Google Scholar, Scopus, Nature index, or WebOfScience. The utilized system has a major advantage over these conventional publication output tools, because it provides an age-normalized ranking. To our knowledge, our system is the only one providing age-normalized rankings. (We would be happy if the reviewer can help us showing countries with similar systems or tools.) 

We have extended the introduction with more information about the ranking.

> They should also criticaly discuss their findings in the disscusion section. 

A new paragraph was added to the Discussion describing limitations of the tool. 

In addition, we already had two other paragraphs about the use of such a tool in decision making. Starting with: 

“A common criticism of such a system focusing on numerical publication output is that here quantity is put before quality…”, and

“With this in mind one could speculate about employing a grant selection solely on the basis of scientometric parameters…”

We have not added further extension for these issues, as at these locations we already provide detailed discussion.

> The authors shoud also focus more on the question if the review process improved after the tool introduction

We asked the grant reviewers whether the provided ranking was useful when evaluating the applicants. To this question, the majority of the reviewers selected that the ranking was useful (Figure 5). We did not ask for specification about the usefulness (e.g. faster or easier review process). We will add this dimension in the future, once we make an evaluation using a longer time period of grant review.

---

## [Decision Letter · Decision Letter 1]

4 Dec 2022

PONE-D-22-24996R1Supporting grant reviewers through the scientometric ranking of applicantsPLOS ONE

Dear Dr. Győrffy,

Thank you for submitting your manuscript to PLOS ONE. After careful consideration, we feel that it has merit but does not fully meet PLOS ONE’s publication criteria as it currently stands. Therefore, we invite you to submit a revised version of the manuscript that addresses the points raised during the review process.

We look forward to receiving your revised manuscript.

Kind regards,

Fausto Cavallaro, PhD

Academic Editor

PLOS ONE

Reviewers' comments:

Reviewer's Responses to Questions

**Comments to the Author**

1. If the authors have adequately addressed your comments raised in a previous round of review and you feel that this manuscript is now acceptable for publication, you may indicate that here to bypass the “Comments to the Author” section, enter your conflict of interest statement in the “Confidential to Editor” section, and submit your "Accept" recommendation.

Reviewer #1: All comments have been addressed

Reviewer #3: (No Response)

2. Is the manuscript technically sound, and do the data support the conclusions?

Reviewer #1: Yes

Reviewer #3: Partly

3. Has the statistical analysis been performed appropriately and rigorously? 

Reviewer #1: Yes

Reviewer #3: I Don't Know

4. Have the authors made all data underlying the findings in their manuscript fully available?

Reviewer #1: Yes

Reviewer #3: No

5. Is the manuscript presented in an intelligible fashion and written in standard English?

Reviewer #1: Yes

Reviewer #3: Yes

6. Review Comments to the Author

Reviewer #1: Thanks for addressing my comments not sure why I need to write 100 characters here change this feature

Reviewer #3: The manuscript "Supporting grant reviewers through the scientometric ranking of applicants" presents the Hungarian portal www.scientometrics.org and its usage in grant decisions. The manuscript is very descriptive, maybe too desciptive.

Bornmann and Daniel (2007) proposed to calculate type I and type II errors for measuring (dis-) agreement between bibliometric values and peer review decisions. Such calculation of type I and type II errors would be interesting for this dataset.

The authors mention that the portal www.scientometrics.org presents discipline- and time- normalized indicators. The details of the normalization procedure are not presented. Which discipline delineation was used? How were tied researchers handled? How large (in terms of number of researchers) are the groups of discipline and time groups? How do the distributions of the values look like? Normalizations are only useful if the entitied to be normalized are numerous enough. For percentiles and deciles, it is additionally important to have a rather well distribution of the values.

From Figure 2c I think that handling of ties is problematic: More than two thirds of the researchers in this example discipline and age seem to have not a single first/last Q1 article. Why is this example researcher positioned around rank 900? All researchers between ranks 300 and 1500 seem to have not a single first/last Q1 article. This researcher seems to be in the top deciles in two of the three measures and tied with two third of the other researchers in the third measure. It is strange that the average result is the fourth decile.

The h index has been heavily criticized. Composite scores also have been criticized. The manuscript lacks a thorough discussion of the problems of the indicators employed in the portal www.scientometrics.org.

Some more specific comments are listed below:

- Page 4, section "Publication and citation data": "... the citation data in MTMT is catalogued to list dependent and independent citations for each publication." Usually, dependent citations are referred to as self-citations in the scientometric literature, and independent citations are the citations without self-citations. This terminology is used on page 9 and should be used throghout the manuscript. Note that not all self-citations are illigitemate. A more open discussion should be provided.

- Page 4, section "Publication and citation data": "To increase simplicity, the average of the three percentiles is computed (in this, the number of publications has a double weight), and this final score is transformed into deciles between D1 and D10, where D1 is the best." Why was the average of the percentiles used? Usually, the median would be used.

The manuscript should be proofread more carefully, see for example:

- Page 4, section "Publication and citation data": "The number of publications is determined differentially in each scientific discipline ... ."  "The number of publications is determined differently in each scientific discipline ... ."

- Page 5, section "Reviewer feedback": "In additional to general data ... ."  "In addition to general data ... ."

- Page 5, section "Statistical analysis": "... submitted and founded applications."  "... submitted and funded applications."

- Page 10: "... as least ..."  "... at least ..."

- Page 10: "In science, the citations and journal rankings thus have similar role to that of the GDP-figures when discussing the country’s economic performance."  "In science, the citations and journal rankings thus have a similar role to that of the GDP-figures when discussing the country’s economic performance."

- Page 10, plural-singular issue: "With the easily available ranking, a higher proportions of grant applicants with higher score received funding."

L. Bornmann & H.-D. Daniel, Convergent validation of peer review decisions using the h index: Extent of and reasons for type I and type II errors, Journal of Informetrics 1(3), 204-213, DOI: 10.1016/j.joi.2007.01.002

7. PLOS authors have the option to publish the peer review history of their article (what does this mean?). If published, this will include your full peer review and any attached files.

Reviewer #1: No

Reviewer #3: No

---

## [Author Response · Author response to Decision Letter 1]

9 Dec 2022

>Reviewer #1: Thanks for addressing my comments not sure why I need to write 100 characters here change this feature

We thank Reviewer #1 for the positive acceptance of our previous edits.

>Reviewer #3: The manuscript "Supporting grant reviewers through the scientometric ranking of applicants" presents the Hungarian portal www.scientometrics.org and its usage in grant decisions. The manuscript is very descriptive, maybe too desciptive. Bornmann and Daniel (2007) proposed to calculate type I and type II errors for measuring (dis-) agreement between bibliometric values and peer review decisions. Such calculation of type I and type II errors would be interesting for this dataset.

L. Bornmann & H.-D. Daniel, Convergent validation of peer review decisions using the h index: Extent of and reasons for type I and type II errors, Journal of Informetrics 1(3), 204-213, DOI: 10.1016/j.joi.2007.01.002

This is an excellent suggestion, and we have now computed both type I and type II errors for all applicants. The Abstract, the Methods, and the Results sections were extended and a new Table was added to the manuscript with the results. The suggested reference is cited in both the manuscript text and at the table description.

>The authors mention that the portal www.scientometrics.org presents discipline- and time- normalized indicators. The details of the normalization procedure are not presented. Which discipline delineation was used? How were tied researchers handled? How large (in terms of number of researchers) are the groups of discipline and time groups? How do the distributions of the values look like? Normalizations are only useful if the entitied to be normalized are numerous enough. For percentiles and deciles, it is additionally important to have a rather well distribution of the values.

We agree with the reviewer and have now extended the manuscript at multiple locations. The disciplines were set up using the classification of the Hungarian Academy of Sciences. We added a link to a publication where we describe the disciplines in detail. Ties were handled by using the median rank for each included researcher. We have added two new figures (Figure 2C and Figure 2D) showing the distribution of the researchers in the different scientific disciplines and across the different publication ages. Note that the age distribution is for the first publication, and a researcher is included in the analysis in all subsequent years.

>From Figure 2c I think that handling of ties is problematic: More than two thirds of the researchers in this example discipline and age seem to have not a single first/last Q1 article. Why is this example researcher positioned around rank 900? All researchers between ranks 300 and 1500 seem to have not a single first/last Q1 article. This researcher seems to be in the top deciles in two of the three measures and tied with two third of the other researchers in the third measure. It is strange that the average result is the fourth decile.

Yes, this researcher is in the top deciles in two categories but had no papers in the last five years. Therefore, the overall rank will not be in the top decile. On the online portal, ties are handled by using the median rank for each included researcher. We extended the manuscript to increase clarity.

>The h index has been heavily criticized. Composite scores also have been criticized. The manuscript lacks a thorough discussion of the problems of the indicators employed in the portal www.scientometrics.org.

We added additional text to the discussion about the limitations of the H-index. In our previous publications (see last paragraph of the introduction and references #3, #4, and #5) we have already extensively analyzed the benefits, problems, and advantages of the different scientometric parameters in different cohorts of researchers. Here, our aim was not to analyze and discuss the indicators used by the portal but to discuss the utility of a decision support tool for grant reviewers. Therefore, here we have not repeated our previous discussions.

>Some more specific comments are listed below:

- Page 4, section "Publication and citation data": "... the citation data in MTMT is catalogued to list dependent and independent citations for each publication." Usually, dependent citations are referred to as self-citations in the scientometric literature, and independent citations are the citations without self-citations. This terminology is used on page 9 and should be used throghout the manuscript. Note that not all self-citations are illigitemate. A more open discussion should be provided.

We have edited the manuscript to adhere to a uniform terminology. Furthermore, the Discussion now also elaborates on the issue of the self-citations.

>- Page 4, section "Publication and citation data": "To increase simplicity, the average of the three percentiles is computed (in this, the number of publications has a double weight), and this final score is transformed into deciles between D1 and D10, where D1 is the best." Why was the average of the percentiles used? Usually, the median would be used.

We have only used three parameters, and the H-index and the yearly citation count are both based on the number of citations. Therefore, these two parameters are closely related. If we used the median value, it would result in using the H-index or the citation rank in most cases. We used the average to avoid this “overfitting to citation count”.

>The manuscript should be proofread more carefully, see for example:

- Page 4, section "Publication and citation data": "The number of publications is determined differentially in each scientific discipline ... ."  "The number of publications is determined differently in each scientific discipline ... ."

- Page 5, section "Reviewer feedback": "In additional to general data ... ."  "In addition to general data ... ."

- Page 5, section "Statistical analysis": "... submitted and founded applications."  "... submitted and funded applications."

- Page 10: "... as least ..."  "... at least ..."

- Page 10: "In science, the citations and journal rankings thus have similar role to that of the GDP-figures when discussing the country’s economic performance."  "In science, the citations and journal rankings thus have a similar role to that of the GDP-figures when discussing the country’s economic performance."

- Page 10, plural-singular issue: "With the easily available ranking, a higher proportions of grant applicants with higher score received funding."

We thank the reviewer for notifying these errors, and we have re-checked and corrected the manuscript at multiple locations.

---

## [Decision Letter · Decision Letter 2]

3 Jan 2023

Supporting grant reviewers through the scientometric ranking of applicants

PONE-D-22-24996R2

Dear Dr. Győrffy,

We’re pleased to inform you that your manuscript has been judged scientifically suitable for publication and will be formally accepted for publication once it meets all outstanding technical requirements.

Kind regards,

Fausto Cavallaro, PhD

Academic Editor

PLOS ONE

Additional Editor Comments: On the base ot the reviewers comments the paper can be accepted.

Reviewers' comments:

Reviewer's Responses to Questions

**Comments to the Author**

1. If the authors have adequately addressed your comments raised in a previous round of review and you feel that this manuscript is now acceptable for publication, you may indicate that here to bypass the “Comments to the Author” section, enter your conflict of interest statement in the “Confidential to Editor” section, and submit your "Accept" recommendation.

Reviewer #3: All comments have been addressed

2. Is the manuscript technically sound, and do the data support the conclusions?

Reviewer #3: Yes

3. Has the statistical analysis been performed appropriately and rigorously? 

Reviewer #3: Yes

4. Have the authors made all data underlying the findings in their manuscript fully available?

Reviewer #3: Yes

5. Is the manuscript presented in an intelligible fashion and written in standard English?

Reviewer #3: Yes

6. Review Comments to the Author

Reviewer #3: The authors have addressed my comments appropriately. Stuff to complete the 100 characters minimum.

7. PLOS authors have the option to publish the peer review history of their article (what does this mean?). If published, this will include your full peer review and any attached files.

Reviewer #3: No

---

## [Editor Report · Acceptance letter]

11 Jan 2023

PONE-D-22-24996R2 

Supporting grant reviewers through the scientometric ranking of applicants 

Dear Dr. Győrffy:

I'm pleased to inform you that your manuscript has been deemed suitable for publication in PLOS ONE. Congratulations! Your manuscript is now with our production department. 

Kind regards, 

on behalf of

Professor Fausto Cavallaro 

Academic Editor

PLOS ONE